# Health-Related Quality of Life in Oral Cancer Patients: Scoping Review and Critical Appraisal of Investigated Determinants

**DOI:** 10.3390/cancers13174398

**Published:** 2021-08-31

**Authors:** Davide De Cicco, Gianpaolo Tartaro, Fortunato Ciardiello, Morena Fasano, Raffaele Rauso, Francesca Fiore, Chiara Spuntarelli, Antonio Troiano, Giorgio Lo Giudice, Giuseppe Colella

**Affiliations:** 1Department of Neurosciences, Reproductive and Odontostomatological Sciences, University of Naples “Federico II”, 80131 Naples, Italy; davide.decicco@unina.it (D.D.C.); chiara.spuntarelli@unina.it (C.S.); giorgio.logiudice@gmail.com (G.L.G.); 2Department of Multidisciplinary Medical, Surgical and Dental Specialties, University of Campania “Luigi Vanvitelli”, 80138 Naples, Italy; gianpaolo.tartaro@unicampania.it (G.T.); raffaele.rauso@unicampania.it (R.R.); giuseppe.colella@unicampania.it (G.C.); 3Department of Precision Medicine, University of Campania “Luigi Vanvitelli”, 80138 Naples, Italy; fortunato.ciardiello@unicampania.it (F.C.); morena.fasano@unicampania.it (M.F.); 4Department of Internal and Polyspecialist Medicine, A.O.U. “Luigi Vanvitelli”, 80131 Naples, Italy; psico.oncologia@unicampania.it

**Keywords:** quality of life, HRQOL, head, neck, oral cancer, oncology

## Abstract

**Simple Summary:**

Oral cancer may strongly impair patients’ quality of life. Huge efforts have been made during recent decades in trying to improve the treatment outcomes in terms of patients’ survival, self-perception, and satisfaction. Consequently, the investigation into health-related quality of life (HRQOL) became an established and worldwide practice. Hundreds of studies tried to clarify which could be the most important variables that impact HRQOL in head and neck cancer patients. However, such a complex topic may be influenced by a multitude of interconnected aspects and several controversies were reported. In this study the current literature was reviewed to identify all those possible sources of bias that may be encountered in trying to correlate HRQOL to patient-specific or disease/treatment-specific aspects. As a result, a list of recommendations was reported to enhance the evidence of future studies.

**Abstract:**

Background: health-related quality of life (HRQOL) represents a secondary endpoint of medical interventions in oncological patients. Our aim was to highlight potential sources of bias that could be encountered when evaluating HRQOL in oral cancer patients. Methods: this review followed PRISMA-ScR recommendations. Participants: patients treated for oral cancer. Concept: HRQOL assessed by EORTC QLQ-C30 and QLQ-H&N35/QLQ-H&N43. A critical appraisal of included studies was performed to evaluate the accuracy of data stratification with respect to HRQOL determinants. Results: overall, 30 studies met the inclusion criteria, totaling 1833 patients. In total, 8 sociodemographic (SDG) and 15 disease/treatment-specific (DT) HRQOL determinants (independent variables) were identified. The mean number of the independent variables was 6.1 (SD, 4.3)—5.0 (SD, 4.0) DT-related and 1.1 (SD, 1.8) SDG-related variables per article. None of the included papers considered all the identified determinants simultaneously. Conclusions: a substantial lack of evidence regarding HRQOL determinants was demonstrated. This strongly weakens the reliability of the reported findings due to the challenging presence of baseline confounding, selection, and omitted variable biases. The proposed approach recommends the use of further evaluation tools that gather more variables in a single score together with a selection of more homogeneous, reproducible, and comparable cohorts based on the identified baseline confounding.

## 1. Introduction

Patient-reported outcomes (PROs) provide precious information about troubles in everyday life and the perception of psychological and physical wellness from the patient’s perspective. Over recent decades, PROs have gained more relevance in treatment decision making, so much so that the U.S. Food and Drug Administration (FDA) and the European Medicines Agency (EMA) consider them—including the quality of life—as a relevant end point to approve new therapies [1,2,3]. To approach such a complex topic as the quality of life in oncological patients, they commonly refer to health-related quality of life (HRQOL). A distinction between these concepts has been made to exclude influences from domains that are not related to the patient’s health status [4], at least theoretically.

The concept of “quality of life” was firstly introduced by Heckscher [5], and in 1977 was adopted as a “keyword” by the United States National Library of Medicine [6]. Since then, several definitions have been proposed [7,8]. The WHO defined quality of life as “individuals perceptions of their position in context of the culture and value systems in which they live and in relation to their goal, expectations, standards, and concerns” [9].

Head and neck tumors and their treatment may negatively affect patients’ HRQOL, which is considered an essential secondary outcome of treatment nowadays [10,11]. For this reason, having reliable evaluation tests is mandatory to better understand how and why specific medical interventions should be chosen and adapted according to individual needs. The quest towards the perfect quality of life evaluation test led researchers to understand some key points to be focused on: a test should be reproducible, sensitive, and easy to understand [12]. Questionnaires developed by the European Organization for Research and Treatment of Cancer (EORTC) Quality of Life Group are widely used in current literature to address these needs. A core questionnaire (EORTC QLQ-C30) is associated with site-specific validated modules (EORTC QLQ-H&N35/43), consisting of single- and multi-item scales that measure several head and neck symptoms [13,14].

HRQOL is a complex topic and needs to be analyzed taking into account every potential influencing factor. Various sociodemographic, disease-specific, and treatment-specific aspects have been recognized as affecting HRQOL [12,15,16,17,18,19,20]. Several researchers have investigated its intrinsic multidimensionality, concluding that HRQOL plays a role in treatment decision making, but none have verified what the relevant items are and how this feature is assessed. The scope of the present review was to highlight possible sources of bias that could be encountered when evaluating HRQOL in patients treated for oral cancer. The second aim was to lay the foundation of a standardized protocol for cohort selection, data collection, and stratification that could enhance knowledge in the field.

## 2. Materials and Methods

This study was conducted following recommendations by PRISMA for scoping reviews (see Appendix A). Description of primary objectives was carried out according to the JBI reviewer’s manual [21]: participants = patients treated for oral cancer; concept = HRQOL assessed by EORTC questionnaires; context = not specified).

A systematic search of published literature was performed in PubMed, EMBASE, and Scopus databases without limitations concerning the date of publication (last screening on 2 February 2021), based on the following search query: (oral cancer OR oral cancers OR tongue cancer OR tongue cancers OR mandible cancer OR cancer of floor of the mouth OR cancers of floor of the mouth OR fom cancer OR fom cancers OR palate cancer OR palate cancers OR palatal cancer OR palatal cancers OR cheek cancer OR cheek cancers OR buccal cancer OR buccal cancers OR gingival cancer OR gingival cancers) AND (quality of life OR health-related quality of life OR health related quality of life OR hrqol OR qol) AND eortc.

All results were exported to Endnote™ bibliographic management software (Clarivate™, Philadelphia, PA, USA). After duplicates removal, the study design filter was applied according to the inclusion/exclusion criteria reported in Table 1. To minimize potential language selection biases, all non-English language papers were moved to the title and abstract screening phase if at least the abstract was reported in the English language. Two authors (D.D.C. and C.S.) independently screened retrieved articles by titles and abstracts. Eventual controversies were solved by the intervention of a third author (G.C.). Those papers considered relevant for the topic were selected for full-text reading and independently screened by two authors (D.D.C. and C.S.) following inclusion/exclusion criteria reported in Table 1. Disagreements were solved by a third author (G.C.). The PRISMA search flow diagram reported in Figure 1 summarizes our strategy.

### 2.1. Data Extraction

According to the findings reported in screened studies and previously published reviews [12,16,20], those sociodemographic (SDG) and disease/treatment-specific (DT) variables that have been found to be linked to patients’ HRQOL were identified and listed.

The following information was retrieved from included studies: country; study design; characteristics of studied populations, such as sample size; SDG features—gender, age, marital status/family, comorbidity, smoke addiction, alcohol consumption, educational level, employment status; DT features—tumor site, tumor T stage, mandibular resection, extent of resection, surgical approach, neck dissection (ND), reconstruction, neoadjuvant radiotherapy (nRT) and adjuvant radiotherapy (RT), neoadjuvant chemotherapy (nCT) and adjuvant chemotherapy (CT), neoadjuvant chemoradiotherapy (nCRT) and adjuvant chemoradiotherapy (CRT), presence of synchronous lesions at baseline, recurrence or metachronous lesions developed before HRQOL evaluation, major postsurgical complications occurred, secondary surgery required. Additional information was retrieved during the appraisal of the included studies (as well as the use of further scoring systems).

### 2.2. Critical Appraisal

Included studies were evaluated and marked as follows:“Stratified” for each independent variable related to EORTC QLQ-C30 and/or EORTC QLQ-H&N35/43 *.“Homogeneous” for each independent variable when all the included cases were equal concerning that specific feature.“Excluded” or “not present in the sample” for each independent variable if the cases reporting that specific feature were excluded during cohort selection, or if that specific feature was not observed in the screened population.“Incomplete stratification” for each independent variable related to EORTC QLQ-C30 and/or EORTC QLQ-H&N35/43, in case of uneven or incomplete sample grouping rules.“Not stratified” for each independent variable that was reported but not related to EORTC QLQ-C30 and/or EORTC QLQ-H&N35/43.“Not available” for each independent variable that did not clearly describe or was not described in the sample features.

A color-coding system was applied as follows:GREEN: stratified, stratified by oral subsites, homogeneous, excluded, not present in the sample.YELLOW: incomplete stratification, incomplete stratification by oral subsites.LIGHT RED: not stratified, not stratified by oral subsites.RED: not available, not clear.

Included articles and independent variables were systematized and charted by using Microsoft^®^ Excel^®^ (v 2012, © 2021 Microsoft Corporation, Albuquerque, NM, USA).

* Specifically, for tumor site, “stratified” and “not stratified” were replaced by “stratified by oral subsites” and “not stratified by oral subsites”, respectively, given that differences were found among tumors located in different oral subsites about their influence on patients’ HRQOL.

## 3. Results

The initial search yielded a total of 1655 studies. Firstly, 403 duplicated records were removed. Then, in accordance with the applied study design criteria (Table 1), 547 records were excluded (488 conference abstracts, 1 conference review, 47 reviews, 6 books, 2 book chapters, 1 editorial, 2 short surveys). The remaining 705 records were screened by title and abstracts (including 37 non-English language papers), resulting in 223 articles that were considered relevant for the topic and selected for full-text reading. The online search finally yielded 25 articles that met inclusion/exclusion criteria. The screening of grey literature and citations of included studies revealed 5 more relevant papers. Thus, a total of 30 studies was included for the critical appraisal. The search strategy is summarized in the PRISMA flow diagram (Figure 1). Although outside the scope of the adopted study design, reasons for the exclusion after full-text reading are summarized in Figure 2 and extensively reported in the Appendix A. The most common reason for exclusion was related to the heterogeneity of the studied cohorts (or poor data stratification) regarding the tumor location.

According to previously published reviews [12,16,20] and included articles, we identified and drafted 23 potential determinants of HRQOL (see Appendix A). Almost all of them were considered as an independent variable for statistical analysis by at least one of the included studies, except for employment status, which was elsewhere advocated to influence HRQOL [22,23].

### 3.1. Study Design

A summary of data design, overall data stratification, and findings of included studies is reported in Table 2. In total, 18 were cohort studies (15 prospective and 3 retrospective), 11 followed a cross-sectional design, and 1 was a case-control study. Of the relevant studies, 27 were conducted on a single-center population, three were multicenter studies (one prospective cohort, one retrospective cohort, and one cross-sectional study). The whole sample of this review comprised 1833 OC cases.

### 3.2. Sociodemographic Variables (SDG)

A summary of data stratification by SDG variables is reported in Table 3 (for further features see Appendix A). In total, 8 of the 23 selected variables were related to SDG aspects. None of the included articles considered all SDG variables simultaneously during cohort selection or for data analysis.

Gender was reported by 28 articles, and data stratification was performed by 10 [9,25,26,27,37,43,46,48,49].

Age was reported by 28 articles; data stratification was properly performed by six [9,26,27,46,49] and inadequately by two (which did not report age thresholds) [43,48]. One study investigated a homogeneous population for this variable [36].

Marital status/family was reported by six articles and data stratification was properly performed by four [25,27,33,43].

Comorbidity status was reported by seven articles, of which, data stratification was properly performed by four papers [25,26,33,46]. One study excluded patients affected by severe comorbidity status [27].

Smoking was reported by seven articles and data stratification was performed by one [27].

Alcohol consumption was reported by four articles and data stratification was performed by two [27].

Educational level was reported by four articles and data stratification was performed by two [27,33].

Employment status/annual income was reported by three articles and data stratification was performed by one [33].

### 3.3. Disease- and Treatment-Specific Variables (DT)

Summary of data stratification by DT variables is reported in Table 3 (for further information see Appendix A). In total, 15 of the 23 selected variables were disease- and treatment-related aspects, seven of which were linked to surgical procedures (see methods paragraph).

None of the included articles considered all DT variables simultaneously during cohort selection or data analysis.

Data from the included studies were adequately stratified by involved oral subsites in three papers [37,46,48] and incompletely/inadequately in five (which customarily grouped different oral subsites) [9,23,30,36,49]. Investigations performed by five studies were on homogeneous populations regarding this variable: on mobile tongue cancers in three [28,31,45], on lower lip cancers in one [38], and on buccal mucosa cancers in another [39].

Tumor stage was reported in 26 articles—data stratification was properly performed in three [9,36,46] and incomplete/inadequate stratification was performed in nine (which distinguished patients grouping different T stages together) [23,24,25,26,27,33,34,49,50,51]; in three studies the investigations were performed on homogenous populations regarding this variable: on pT3 of the mobile tongue in two [28,31] and on T4 of the buccal mucosal in the other [39].

Of the included studies, two were conducted on patients who had undergone medical treatments without surgery [45,49], thus marked as “not present” (NP) compared to all the surgery-related DT variables. The only exception was the study of Petruson et al. [45], which was marked as “not available” for “required secondary surgery” since the authors did not clearly define whether a part of the studied sample underwent a secondary surgery after definitive medical treatment.

Performed mandibular resection was overtly reported in 15 articles—data stratification was properly performed in three [23,36,46]; incomplete/inadequate stratification was performed in one (which compared no mandibular resection group to patients undergoing mandibular resection grouping together with those who received marginal and segmental resections) [24]; six studies clearly stated that none of the included cases underwent mandibular resection [28,31,38,39,45,49]; and in three studies, the investigated population homogeneously underwent segmental mandibular resection [29,40,48].

The extent of surgical resection was considered “stratified” only in those cases where the resected oral subsites were clearly identified. This variable was indicated in seven articles—according to this definition, none performed stratifications. Data from one study were considered incompletely/inadequately stratified due to the reported horizontal defect size (which partially defined the extent of surgical resection) [48]. In two studies, the investigated population homogeneously underwent the same resection: partial glossectomy in one [28] and partial pelviglossectomy in the other [31].

The surgical approach was indicated in seven articles; data stratification was properly performed in one [31]. In three studies, the investigated population homogeneously underwent transoral surgery [28,38,39].

The performed ND was indicated in 10 articles—data stratification was properly performed in one (it means that different standardized procedures [52] were separately investigated) [37] and incomplete/inadequate stratification was performed in six (mostly because the type of ND were not specified) [10,23,28,36,39,48].

The performed reconstruction was reported in 23 articles—data stratification was properly performed in five (means that each investigated reconstruction strategy—i.e., each type of free flap, each type of regional flap, each type of local flap, primary closure, and each type of graft was investigated separately from each other) [28,29,32,38,40], incomplete/inadequate stratification was performed in in seven [23,24,31,35,37,46,48], and the investigated populations homogeneously underwent the same reconstruction strategy in four studies (radial forearm free flap) [22,25,26,39].

The performed nRT was reported in 12 articles—data stratification was properly performed by one [35] and incomplete/inadequate stratification was performed by two (means that the authors did not define whether the radiotherapy was performed before or after surgery) [32,33]. In 1 study the investigated population homogeneously underwent nCRT [36]; 5 studies stated that none of the included cases underwent nRT [31,34,39,45,49].

The performed nCT was reported in eight articles—data stratification was properly performed by one [35]; incomplete/inadequate stratification was performed by another one (it means that authors did not define whether the radiotherapy was performed before or after surgery) [33]; in one study, the investigated population homogeneously underwent nCRT [36]; and five articles stated that none of the included cases underwent nCT [31,34,39,45,49].

The performed RT (both adjuvant or definitive) was reported in 26 articles—data stratification was properly performed by 10 [26,27,30,34,35,37,39,41,47,48]; incomplete/inadequate stratification was performed by two (this means that authors did not define whether radiotherapy was performed before or after surgery) [32,33]; and in 4 studies, the investigated population homogeneously underwent RT or CRT (both adjuvant or definitive) [22,28,45,49].

The performed CT (both adjuvant or definitive) was reported in 13 articles—data stratification was properly performed by four [30,34,37,39]; incomplete/inadequate stratification was performed by one (it means that authors did not define whether radiotherapy was performed before or after surgery) [33]; in three studies, the investigated population homogeneously underwent RT or CRT (both adjuvant or definitive) [28,45,49]; and one article overtly stated that none of the included cases underwent CT [35].

The presence or the absence of patients with synchronous lesions at baseline in the studied sample was overtly indicated in four articles—in one paper, those patients who presented synchronous lesions were excluded a priori [28], while the authors in two studies stated that these patients were not present in the studied population [31,49].

**Table 3 cancers-13-04398-t003:** Summary of data stratification by DT and SDG variables. Legend to Table 3: Ex = excluded; H = homogeneous; IS = incomplete/inadequate stratification; ISOS = incomplete/inadequate stratification by oral subsites; na = not available; NP = not present; NS = not stratified; NSOS = not stratified by oral subsites; S = stratified; SOS = stratified by oral subsites. * Studies on patients treated by non-surgical therapies.

		DISEASE AND TREATMENT VARIABLES (DT)						SOCIODEMOGRAPHIC VARIABLES (SDG)						TOTAL VARIABLES CONSIDERED FOR DATA STRATIFICATION (DT+SDG)					
Article	OC Sample	(1) S/SOS/H/EX/NP	(2) IS/ISOS	(3) NS/NSOS	(4) na	Considered Var (Tot 1 + 2)	Ignored Var (Tot 3 + 4)	(1) S/SOS/H/EX/NP	(2) IS/ISOS	(3) NS/NSOS	(4) na	Considered Var (Tot 1 + 2)	Ignored Var (Tot 3 + 4)	(1) S/SOS/H/EX/NP	(2) IS/ISOS	(3) NS/NSOS	(4) na	Considered Var (Tot 1 + 2)	Ignored Var (Tot 3 + 4)
Airoldi 2011 [22]	50	2	0	3	10	2	13	0	0	7	1	0	8	2	0	10	11	2	21
Beck 2017 [24]	45	2	3	1	9	5	10	0	0	2	6	0	8	2	3	3	15	5	18
Becker 2012 [23]	50	1	4	4	6	5	10	0	0	4	4	0	8	1	4	8	10	5	18
Borggreven 2007 [25]	38	1	1	8	5	2	13	3	0	1	4	3	5	4	1	9	9	5	18
Bozec 2009 [26]	21	3	2	3	7	5	10	3	0	0	5	3	5	6	2	3	12	8	15
Bozec 2020 [27]	48	1	1	2	11	2	13	7	0	0	1	7	1	8	1	2	12	9	14
Canis 2016 [28]	40	10	1	1	3	11	4	0	0	4	4	0	8	10	1	5	7	11	12
Crombie 2014 [10]	16	0	1	9	5	1	14	0	0	2	6	0	8	0	1	11	11	1	22
Davudov 2019 [29]	120	2	0	3	10	2	13	0	0	3	5	0	8	2	0	6	15	2	21
Dzioba 2017 [30]	117	2	1	6	6	3	12	0	0	2	6	0	8	2	1	8	12	3	20
Ferri 2020 [31]	55	12	1	1	1	13	2	0	0	2	6	0	8	12	1	3	7	13	10
Girod 2009 [32]	34	2	2	1	10	4	11	0	0	3	5	0	8	2	2	4	15	4	19
Huang 2010 [33]	129	1	5	1	8	6	9	6	0	0	2	6	2	7	5	1	10	12	11
Infante-Cossio 2009 [34]	70	5	1	1	8	6	9	0	0	2	6	0	8	5	1	3	14	6	17
Kessler 2004 [35]	41	5	1	6	3	6	9	0	0	2	6	0	8	5	2	7	9	7	16
Khandelwal 2017 [9]	50	1	1	3	10	2	13	2	0	0	6	2	6	3	1	3	16	4	19
Klug 2002 [36]	67	5	2	1	7	7	8	1	0	1	6	1	7	6	2	2	13	8	15
Kovacs 2015 [37]	110	4	1	3	7	5	10	1	0	1	6	1	7	5	1	4	13	6	17
Lin 2020 [38]	22	5	0	3	7	5	10	0	0	3	5	0	8	5	0	6	12	5	18
Mair 2017 [39]	225	10	1	0	4	11	4	0	0	2	6	0	8	10	1	2	10	11	12
Moubayed 2014 [40]	13	2	0	3	10	2	13	0	0	1	7	0	8	2	0	4	17	2	21
Nordgren 2008 [41]	122	1	0	2	12	1	14	0	0	3	5	0	8	1	0	5	17	1	22
Oates 2008 [42]	47	1	0	2	12	1	14	0	0	0	8	0	8	1	0	2	20	1	22
Oskam 2013 [43]	38	0	1	3	11	1	14	2	1	3	2	3	5	2	2	6	13	4	19
Peisker 2016 [44]	100	0	0	3	12	0	15	0	0	2	6	0	8	0	0	5	18	0	23
Petruson 2005 * [45]	30	11	0	1	3	11	4	0	0	2	6	0	8	11	0	3	9	11	12
Pierre 2014 [46]	37	4	2	3	6	6	9	3	0	0	5	3	5	7	2	3	11	9	14
Schoen 2008 [47]	41	1	0	3	11	1	14	0	0	2	6	0	8	1	0	5	17	1	22
Van Gemert 2015 [48]	37	5	3	1	6	8	7	1	1	0	6	2	6	6	4	1	12	10	13
Yoshimura 2009 * [49]	20	13	2	0	0	15	0	2	0	0	6	2	6	15	2	0	6	17	6
AVG (SD)		3.7 (3.8)	1.2 (1.3)	2.7 (2.2)	7.3 (3.3)	5.0 (4.0)	10.0 (4.0)	1.0 (1.8)	0.1 (0.3)	1.8 (1.6)	5.1 (1.7)	1.1 (1.8)	6.9 (1.8)	4.8 (3.9)	1.3 (1.3)	4.5 (2.8)	12.4 (3.5)	6.1 (4.3)	16.9 (4.3)
WEIGHTED AVG BY OC SAMPLE (SD)		3.8 (3.7)	1.3 (1.4)	2.4 (1.9)	7.5 (3.2)	5.1 (3.8)	9.9 (3.8)	1.0 (1.9)	0.0 (0.2)	1.9 (1.4)	5.1 (1.6)	1.0 (1.9)	7.0 (1.9)	4.8 (3.7)	1.3 (1.4)	4.2 (2.5)	12.6 (3.2)	6.1 (4.2)	16.9 (4.2)

The presence or the absence of patients who developed metachronous neoplasms or disease relapse in the studied sample were indicated in 17 articles—data stratification was properly performed in two [39,46]; in eight papers, those patients who developed a relapse or a metachronous lesion were excluded from data analysis [24,26,34,36,42,48,50,53]; and the authors in another study overtly stated that these patients were not present in the studied population [31].

The presence or the absence of patients who experienced major post-surgical complications in the studied sample was reported in nine articles—data stratification was properly performed in one [32]; incomplete/inadequate stratification was performed in two (it means that an uneven definition of this variable was reported—e.g., partial and total flap loss not distinguished, major surgical complications NOS) [26,46]; in another paper, these patients were excluded from data analysis [48]; and the authors from three studies clearly stated no major post-surgical complications were observed in the investigated sample [28,31,38].

Patients who required secondary surgery for tumor relapse or reported major post-surgical complications were included in six articles—none performed data stratification regarding this variable; in one study, these patients were excluded from the data analysis [24]; and the authors from another study stated that no secondary surgery was performed in the investigated sample [31].

### 3.4. Descriptive Analysis

RT and gender were the most frequently considered among DT and SDG variables, respectively, followed by mandibular resection and reconstruction in the former group, and by age and comorbidity in the latter (Figure 3 and Figure 4). Results also showed that these studies focused on the exclusion of patients who developed recurrences of metachronous lesions.

On average, only 5.0 (SD, 4.0) DT variables were considered by each included study, and 5.1 (SD, 3.8) for each case, as a result in the weighted average. However, these values dropped to 3.7 (SD, 3.8) and 3.8 (SD, 3.7) if just proper analysis, exclusions, and homogeneity were considered (Table 3, Figure 5).

On average, only 1.1 (SD, 1.8) SDG variables were considered by each included study, and 1.0 (SD, 1.9) for each case, as a result in the weighted average. Similar values were achieved considering only proper analysis, exclusions, and homogeneity (Table 3, Figure 5).

As mentioned above, surgery-related DT variables were considered as “not present” (NP) for those studies that investigated a non-surgical population [45,49]. Thus, they resulted in two of the most accurate analyses among the included studies (Figure 6).

## 4. Discussion

Although this article was initially designed as a systematic review and a meta-analysis, in our opinion, outcomes would be meaningless due to the inhomogeneity of included studies and biases that might have occurred. As a result, we chose to investigate how closely potential influencing factors were evaluated, to highlight possible sources of bias that could be encountered assessing HRQOL in oral cancer patients.

### 4.1. Sociodemographic Variables

#### 4.1.1. Gender and Age

Among sociodemographic variables, gender and age were the most investigated ones. Most of the included studies found no differences concerning these variables [9,25,27,46,49,53]. Remarkably, Kovacs et al. [37] reported worse results in males regarding financial difficulties and cognitive and social functioning. This revealed an interesting food for thought, considering that household income derives most commonly from men.

Non-standardized thresholds were considered by investigating the potential influences of age. Moreover, it is noteworthy that during the last decades chronological age has progressively lost its relevance according to the comprehensive geriatric assessment (GCA) approach. An innovative concept of “psychological age” is gaining momentum in the field [54,55] and it was adapted to HNC patients by Pottel et al. [56], assessing the effectiveness of different health status screening tools. They found that Geriatric 8 (G8) represents the index of choice to identify patients in a GCA approach. Among included studies, Bozec et al. [27] performed a stratification using the G8 tool, finding a significant negative correlation between HRQOL and scores lower than 15.

#### 4.1.2. Marital Status and Family

Marital status and family were investigated by four studies [25,27,33,51], all reporting no associations with questionnaires. However, Bozec et al. [27] found a negative correlation by stratifying the results of the EORTC QLQ-ELD14. This finding suggests the existence of covering effects from other variables that might impact QLQ-C30 and H&N35 strongly, hiding possible influences of the marital status and family conditions.

#### 4.1.3. Comorbidity

Only a minority of the included studies investigated the influence of comorbidity on HRQOL. No correlations were found by three studies [25,33,46], while results from Bozec et al. [26] were retrieved from the analysis of questionnaires taken 6 months after surgery. As reported in the inclusion/exclusion criteria, these suggestions were not taken into account, since a great variability in HRQOL scores was reported in the literature during the first year after treatment.

It must be noted that different methods (even non-standardized and non-validated) were used to assess patients’ comorbidity status. In our opinion, it is strongly preferable to use one of the several scoring systems and scales widely adopted elsewhere in the literature, as well as the Kaplan–Feinstein Index (KFI)—which was developed to evaluate comorbidities in diabetes mellitus [57] and subsequently modified and validated by Piccirillo [58]—or the Adult Comorbidity Evaluation 27 (ACE-27) [59]—which also includes alcohol abuse.

#### 4.1.4. Alcohol, Smoke, and Educational Level

The effects of alcohol consumption and smoking were investigated only by Bozec et al. [27], who found no correlations with HRQOL conversely to other findings reported in the literature [60,61]. To clarify the roles of smoking and alcohol consumption in determining HRQOL, the comparison to some control groups composed of teetotalers and non-smokers should be required. Unfortunately, it would be extremely challenging to obtain adequate sample sizes to allow them to be reliably compared. Smoking and alcohol intake are the main risk factors for the development of OCC [62].

The correlations between HRQOL and educational level were analyzed by Bozec et al. [27] and Huang et al. [33], both retrieving no associations.

Interesting results would be expected from an investigation into educational level in larger cohorts. In this regard, it might be preferred to achieve standardized subgroups by using validated evaluation tools, as well as the International Standard Classification of Education (ISCED) [22].

### 4.2. Disease- and Treatment-Specific Variables

#### 4.2.1. Cancer Site

Although HRQOL in HNC patients has gained great relevance during recent decades, most published studies still have not considered that cancer site might have a significant impact [34,37,42,46,50,63]. Indeed, the most common reason for exclusion in the screened articles was directly related to this aspect (Figure 2). This potential source of bias is scantly contemplated concerning the HNC regions (e.g., oral cavity, oropharynx, larynx, etc.), and much less considering oral subsites.

Interestingly, findings reported by Kovacs et al. [37] demonstrated that cancers arising from different oral subsites differently affect HRQOL, while Pierre et al. [46] and van Gemert et al. [48] found no significant variations. Such controversies should be addressed by analyzing larger samples that allow performing a more reliable data stratification. At the same time, it must be highlighted that the cohort selection would overcome this issue by including more homogeneous cases, as performed by some included articles [28,31,38,39,45].

#### 4.2.2. Cancer Stage

Most likely, the cancer stage represents one of the most challenging variables to correlate with HRQOL, since the multitude of baseline confounding must be considered. For example, compared to early-stage cancers, advanced stages require more frequently adjuvant therapies and they need more extensive surgeries, which may include a mandibular resection, implying more demanding reconstruction strategies. The appraisal of findings reported by Beck-Broichsitter et al. [24] and Becker et al. [23] provided a clear demonstration of possible controversies that could be encountered due to some omitted variable biases. The authors compared the same T-stage subgroups (Tis-2 vs. T3/4) and one found no significant differences, while the other reported worse results for almost all questionnaire items in advanced-stage cancers. Controversies like this are repeatedly presented in the screened papers, some reporting no differences [25,27,36,49], others reporting substantial ones [9,33,34,46].

In our opinion, the cancer stage could be considered in a wider context, including almost all baseline confounding. The only exception is represented by those middle-stage cancers that could or could not be eligible for adjuvant therapies based on clinical and histological features. Future studies will provide adequate piece of evidence to reliably correlate these variables.

#### 4.2.3. Mandibular Resection

Although it has been previously stressed that mandibular resection strongly impairs patients’ HRQOL [16,64], the generic findings of included studies are inconsistent. Becker et al. [23] were the sole researchers reporting worse results in patients undergoing mandibular resection compared to those who did not. A mandibular resection group was also studied distinguishing marginal from segmental resections. Unsurprisingly, the former demonstrated better questionnaire results.

Like most of the selected variables, the controversies observed among included articles suggest the existence of baseline confounding. We suppose that the need for adjuvant therapies (particularly the RT), the reconstruction, the cancer stage, and the extent of surgical resection could be the most probable sources of bias, since the mandible involvement is commonly associated with advanced cancer stages.

#### 4.2.4. Extent of Resection

Van Gemert et al. [48] were the sole researchers who stratified the studied sample according to the extent of resection (specifically in the horizontal size). Conversely to what was documented elsewhere [16,65], they reported minimal differences. Since the current knowledge in reconstructive techniques allows surgeons to adequately restore even complex and extended defects, the authors suggest that accurate and successful reconstructions could justify these findings. We agree with this hypothesis, despite the fact that surgical complications and secondary surgery must be excluded or carefully examined during data analysis to ensure the absence of possible omitted variable biases. Thus, influences from the aforementioned baseline confounding (see cancer stage paragraph) should be considered.

#### 4.2.5. Surgical Approach

Within the included studies, the impact of the surgical approach on HRQOL was supposed to explain some of the findings. Ferri et al. [31] were the only ones who considered this variable for data analysis. They compared two different treatment protocols: transoral partial pelviglossectomy followed by a buccinator artery myomucosal flap versus a pull-through partial pelviglossectomy followed by various free flaps. Significantly better results were reported in the former group.

The comparison of different surgical protocols implies taking into account some baseline confounding. For example, the pull-through resection involves various deep structures of the mouth floor that can more likely be restored by using free flaps [66], as clearly recognized by the authors. The cancer stage, adjuvant therapies, and the extent of resection also represent possible baseline confounding variables, since the cancer extent might force the surgeon to choose more invasive surgical approaches.

As reported elsewhere in the literature, the surgical approach seems to impact HRQOL in treated patients. Although disease-free survival still represents the primary outcome, minimally invasive approaches should be considered whenever it is possible, in order to reduce post-operative morbidity [67,68,69,70,71].

#### 4.2.6. Neck Dissection

Some contradictory results were retrieved from the included studies concerning the ND as an HRQOL determinant. Kovacs et al. [37] described progressively worse results comparing patients who did not receive ND to those treated by selective ND (lev. I–II) and those by type III modified radical ND. We agree with the authors’ opinion about the possibility of baseline confounding since patients undergoing ND most frequently even underwent adjuvant RT. Future studies comparing patients receiving RT/CRT only and those treated by surgery with neck dissection and adjuvant RT/CRT will probably clarify these doubts.

#### 4.2.7. Reconstruction

Unsurprisingly, reconstruction was the most investigated among surgery-related variables. It is commonly believed that the quality of reconstruction is strictly associated with patients’ functional and aesthetic outcomes and post-treatment HRQOL [72,73]. Knowledge in reconstructive surgery has been taking great strides forward since free flaps were introduced for the restoration of head and neck defects [72,73,74]. Performing a systematic review of the literature on reconstructive strategies in patients not eligible for free flaps, we surprisingly highlighted a growing interest toward more conservative solutions over the last few years [75,76,77].

Despite the huge literature, reconstruction still raises disputes about which surgical reconstructive protocol is the best to restore oral defects [78,79,80,81,82,83]. Similarly, the findings reported by included studies showed widely controversial results. In this regard, it should be noticed that huge differences within the studied populations do not permit a reliable comparison of the observed outcomes. In our opinion, the evaluation of the impact of reconstructive procedures on HRQOL implies several risks of bias that must be considered. For instance, careful attention should be paid to patients who developed surgical complications, by excluding them or by performing an accurate sample stratification. Furthermore, the related complications may heavily impair the functional outcome, requiring a much longer recovery time, long-term rehabilitation programs, or even secondary surgery. Moreover, according to the chosen procedure, free flaps may lead to various donor site morbidity [84,85]. All these aspects should be considered for their potential effects on HRQOL.

Reconstruction strategies are mainly chosen according to the defect size and composition: small to moderate simple defects may benefit from reduced donor site morbidity by performing local flaps, while large and/or composite defects need free or regional flaps to be restored [75,86]. Therefore, the evaluation of the reconstruction as an HRQOL determinant should consider some baseline confounding variables, such as the cancer stage, the extent of resection, the mandibular involvement, and the adjuvant therapies. None of the included studies considered simultaneously all these independent variables during the data analysis. Conversely, many of them investigated various reconstructive procedures grouping different flaps together. In our opinion, a reliable comparison should firstly consider the studied flaps separately to minimize evitable biases.

#### 4.2.8. Radiotherapy and Chemotherapy

Almost all the included studies agreed about the deteriorating effects of radiotherapy on HRQOL. Kovacs et al. [37] performed an accurate study comparing patients who received adjuvant RT, adjuvant CT, or adjuvant CRT. Interestingly, there were no significant differences between adjuvant RT and adjuvant CRT groups, which both demonstrated significant worse results compared to patients who did not undergo post-surgical therapies.

Some symptom-related items were found to be particularly affected: dry mouth, sticky saliva, and mouth opening were almost always impaired. These findings were in line with those already widely reported in the published literature [87,88,89,90,91,92,93].

The evaluation of HRQOL demonstrated less interest in studying the effects of neoadjuvant therapies and adjuvant CT alone. This could be attributed to the uncommon use of these treatment protocols in HNC and it would be interesting to investigate the existence of different influences on HRQOL between neoadjuvant therapies and post-surgical ones.

Further compelling aspects derive from the adopted RT technique. The accurate analysis performed by Huang et al. [33] underlined that the most recent 3D radiotherapy (3DRT) and the intensity-modulated radiotherapy (IMRT) result in a better impact on patients’ HRQOL, as largely accepted in the current literature [94,95]. Nevertheless, most included studies did not specify which techniques were used in the studied samples, producing a relevant source of bias.

As mentioned above, adjuvant therapies suffer from several baseline confounding factors that should be always considered during the data appraisal. Nonetheless, the trends in the reported findings overtly suggest that it can be considered as one of the main HRQOL influencing factors.

#### 4.2.9. Synchronous Lesions, Recurrences, and Metachronous Lesions

Although rare, the presence of synchronous lesions in the oral cavity inevitably requires larger resective surgeries that negatively influence the HRQOL, but only three studies clearly excluded these patients [28,31,49].

On the other hand, it might appear obvious that a recurrence of previously treated tumors or the development of further cancers may strongly impair HRQOL, especially by affecting psychological status and symptoms [96,97,98]. Nevertheless, only 12 of the included papers considered this aspect during cohort selection [24,26,31,33,34,35,36,39,42,46,48,49]. Mair et al. [39] were the only ones who conducted an analysis to compare disease-free patients to those who developed a recurrence. Their results strongly support the initial hypothesis, but we should make a point to note the potential sources of bias that might be encountered. Indeed, progression-free survival strongly depends on the cancer stage, which also reflects the invasiveness of the adopted treatment.

#### 4.2.10. Major Surgical Complications and Secondary Surgery

Only a minority of the included studies considered these variables. Girod et al. [32] investigated the differences between the reconstruction of OC defects by using split thickness skin graft and acellular dermal matrix. They stratified the results by surgical complications, distinguishing patients who experienced a graft failure from those with regular healing. No significant differences were found, but the small sample size and the missing stratification by other variables might have affected their results.

It is reasonable to believe that post-surgical complications and recurrences may impact the HRQOL. In our opinion, this might be related to the resulting functional and aesthetic impairments or to the need for secondary surgery, which may impair the psychological status and the symptoms [96,97,98]. The included studies did not investigate this relation and it could be an interesting food of thoughts for future studies.

#### 4.2.11. Other Variables

The increasing knowledge in multidisciplinary management of oncological patients strongly highlights the relevance of the psychological status [99]. The HRQOL is considered useful not only to evaluate the quality of care interventions from the patient’s perspective but also to adjust clinical decision making by evaluating patients’ needs and additional interventions, such as psychological counselling [100]. The close relationship between psychological status and HRQOL was demonstrated to predict the quality of life in patients treated for HNC [101].

Expressions of poor psychological status were investigated among the included studies. Moubayed et al. [40] and Bozec et al. [27] observed a negative correlation between depression and HRQOL, as measured by using the Hospital Anxiety and Depression Scale (HADS), while Airoldi et al. [22] supported this observation after evaluating associations with the Dische Scale. In our opinion, obtaining information on patients’ psychological status is mandatory to avoid biases that could impair the reported observations. Stratifying results by using validated and standardized indexing systems could address this issue.

Dental restoration represents one of the most interesting fields in searching for treatment-related aspects that could improve the HRQOL in OC patients. Usually, dental status has been already impaired at baseline and not only in those who suffered from cancers involving the jaws. The dental prosthetic restoration (supported or not by implants) could be a deeply influencing factor in patients’ everyday life and HRQOL. The recovery of dental occlusion and a balanced mastication has been demonstrated to influence aesthetic outcomes, social parameters, swallowing and cognitive functions [102,103,104,105,106,107,108,109,110,111,112]. The published literature expressed highly significantly better results in patients undergoing micro-vascular mandibular reconstruction (mostly by using free fibula flap) with following implant-supported dental prosthetic rehabilitation compared to non-rehabilitated patients [108,109,110,111,112]. Unfortunately, most of these articles included non-oncological patients within the investigated population and did not meet the inclusion/exclusion criteria.

A potential limitation of this review could be the exclusion of papers that used other evaluation tools. We chose to select only those studies based on EORTC questionnaires because of the comprehensive insight given by the assessment of general (by the QLQ-C30 module) and specific (by the QLQ-H&N35/43 modules) features, addressing the widespread use of these questionnaires. Further studies could provide a comparison with other tools.

## 5. Conclusions and Recommendations for Future Studies

The number of controversies found in the current literature demonstrates a substantial lack of evidence regarding HRQOL determinants in HNC patients. Therefore, none of the potential influencing variables should be excluded from data analysis based on the authors’ opinion only.

Currently, many of the published articles considered a minority of potential determinants. The data analysis is commonly performed on the basis of each independent variable individually. By approaching such a complex and multidimensional aspect as the HRQOL in this way, the reliability of the reported findings might be strongly weakened due to several selection and omitted variable biases that could be encountered. Since the EORTC Quality of Life Group was founded in 1980, a standardized guideline for cohort selection is still lacking. Thus, the crucial task to avoid the described biases is charged to examiners’ knowledge only.

We strongly believe that almost all the identified determinants should be investigated. This implies that much larger samples and much more data must be collected. At the same time, particular attention should be paid to cohort selection to achieve better comparability among the studies. This scope will probably be attained by creating a shared and standardized online data set.

Considering the complex net of baseline confounding highlighted in this manuscript, a suitable strategy could be the use of further evaluation tools, scales, and indexes that condenses many variables in a single score. In our opinion, the benefits from this approach are twofold: a simplification of data analysis and a minimization of omitted variable biases. In this regard, an interesting investigation was performed by Tribius et al. [113] regarding the influence of sociodemographic variables on HRQOL in HNC patients. This study used an adapted version of a composite social class indicator [114] that considered three different sociodemographic variables (educational level, type of occupation, and household income) to differentiate the socio-economic status as high, moderate, or low. Other examples were reported within the discussion of this review (G8, ACE-27, KFI, HADS), but those were related to sociodemographic and psychological variables. To the best of our knowledge, no scoring systems that condense the selected DT-specific variables have been developed yet. Our recommendation for future research is to consider these features simultaneously, rather than individually, addressing the baseline confounding described above, and to select cohorts that are as homogeneous as possible. An example of this protocol is given by Ferri et al. [31] and Canis et al. [28], who performed some accurate cohort selections resulting in quite a small sample size, but one that was highly homogeneous and reproducible.

As observed by Borggreven et al. [25], patients usually present compromised HRQOL at the baseline, probably due to preexisting impairments related to comorbidity status or cancer diagnosis. We believe that this issue could be addressed by evaluating only the differences between baseline and post-treatment questionnaires in a longitudinal study design, rather than in absolute scores compared to a reference population in a cross-sectional fashion, even though the interquestionnaire analysis may highlight interesting insights [44].

As a result of this approach, more homogeneous, reproducible, and comparable cohorts will be expected, enhancing the level of evidence in the field.

## Figures and Tables

**Figure 1 cancers-13-04398-f001:**
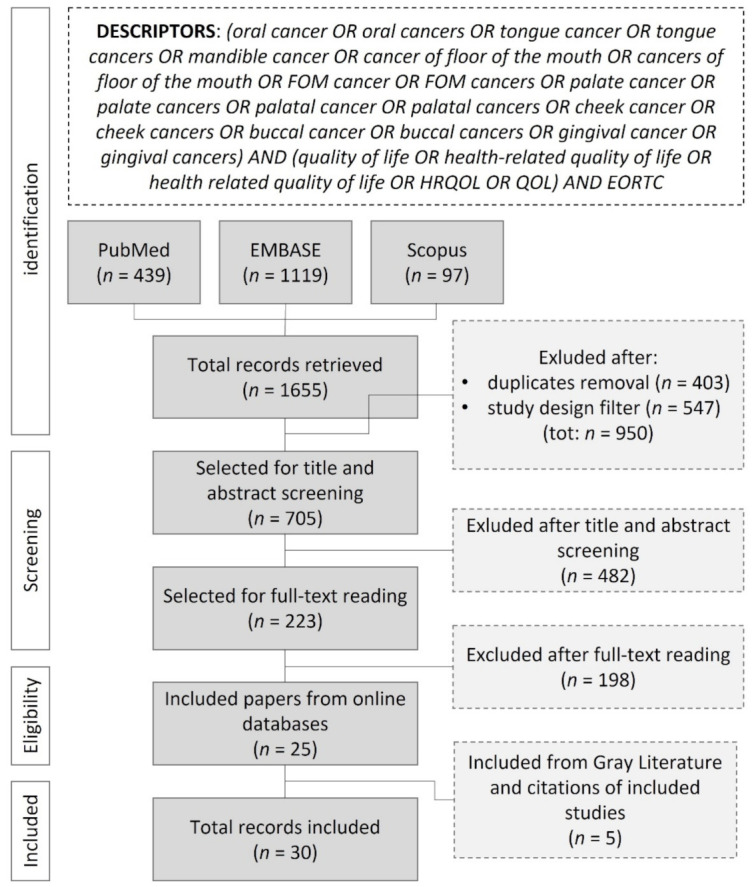
PRISMA search flow diagram.

**Figure 2 cancers-13-04398-f002:**
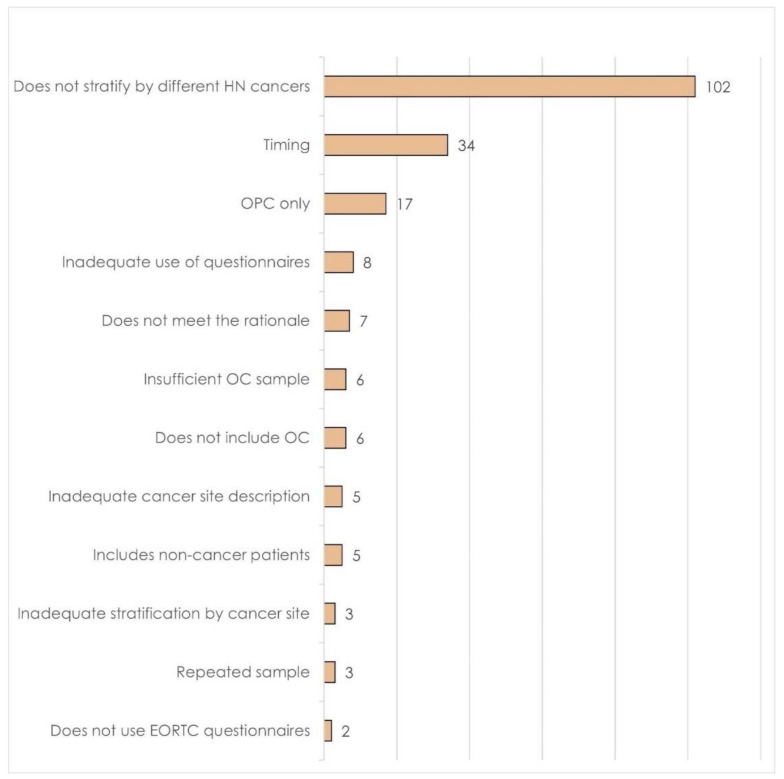
Reasons for the exclusion of the screened articles.

**Figure 3 cancers-13-04398-f003:**
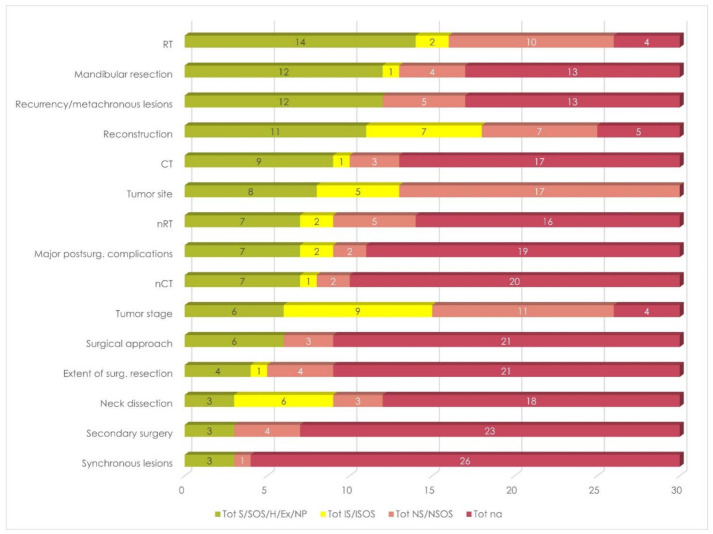
Overall stratification by DT variables. Legend to Figure 3: CT = adjuvant/definitive chemotherapy; Ex = excluded; H = homogeneous; IS = incomplete/inadequate stratification; ISOS = incomplete/inadequate stratification by oral subsites; na = not available; nCT = neoadjuvant chemotherapy; ND = neck dissection; NP = not present; nRT = neoadjuvant radiotherapy; NS = not stratified; NSOS = not stratified by oral subsites; RT = adjuvant/definitive radiotherapy; S = stratified; SOS = stratified by oral subsites.

**Figure 4 cancers-13-04398-f004:**
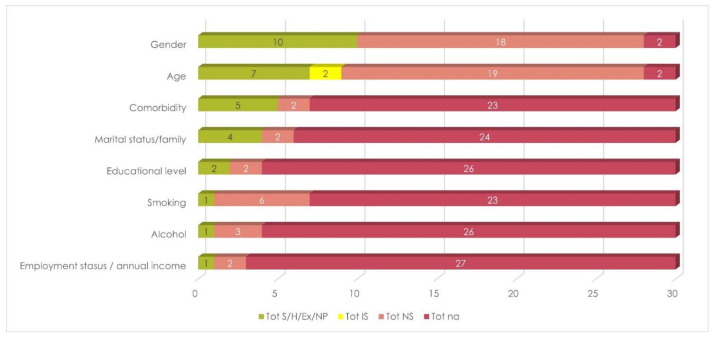
Overall stratification by SDG variables. Legend to Figure 4: Ex = excluded; H = homogeneous; IS = incomplete/inadequate stratification; na = not available; NP = not present; NS = not stratified; S = stratified.

**Figure 5 cancers-13-04398-f005:**
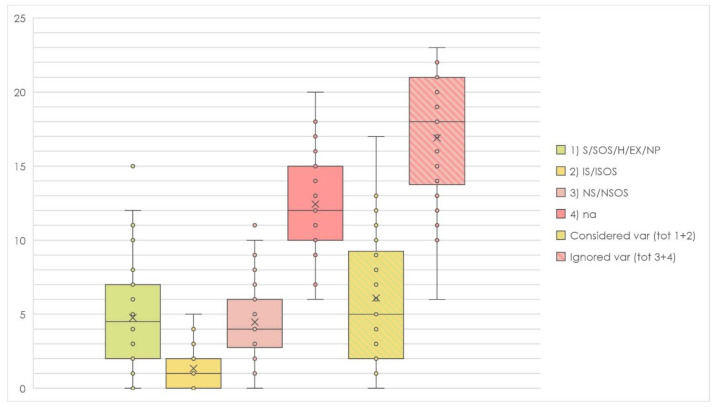
Box plot representation of the considered independent variables during data analysis and cohort selection. Legend to Figure 5: Ex = excluded; H = homogeneous; IS = incomplete/inadequate stratification; ISOS = incomplete/inadequate stratification by oral subsites; na = not available; nc = not clear; NP = not present; NS = not stratified; NSOS = not stratified by oral subsites; S = stratified; SOS = stratified by oral subsites.

**Figure 6 cancers-13-04398-f006:**
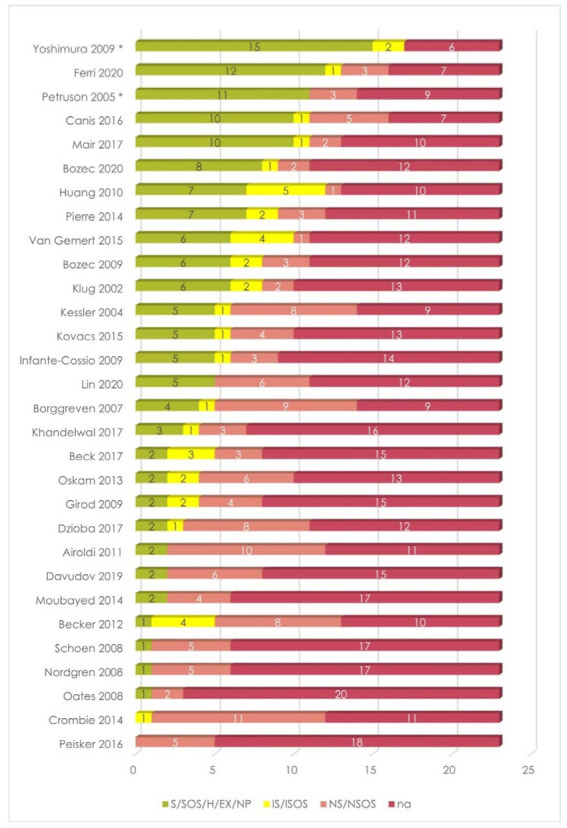
Summary of data stratification by both SDG and DT variables of the included studies. * Studies on patients treated by non-surgical therapies.

**Table 1 cancers-13-04398-t001:** Inclusion/exclusion criteria.

INCLUSION CRITERIA	
Type of study	Randomized/non-randomized trials, cohort studies, cross-sectional, case control, prospective, retrospective studies
Cohort	Patients treated for oral cancer
Sample	≥10
Data	Quality of life assessed by using both EORTC QLQ-C30 and EORTC QLQ-H&N35/43
Timing	Evaluation of HRQOL performed after at least 12 months since treatment
**EXCLUSION CRITERIA**	
Type of study	Case series, case reports, reviews, letters, technical notes, conference documents, books, book chapters, editorials, surveys
Cohort	Studies on patients treated for non-oral/oropharyngeal cancers without stratification. Studies on non-treated patients
Sample	<10
Data	Studies using other HRQOL evaluation tools
Timing	Last HRQOL assessment performed before 12-months post-treatment

**Table 2 cancers-13-04398-t002:** Study design and independent variables considered for data stratification and findings. Legend to Table 2: ACE-27 = Adult Comorbidity Evaluation 27 score; ADM = acellular dermal matrix; BAMM = buccinator myomucosal flap; BOT = base of tongue; CRT = chemoradiotherapy; CT = chemotherapy; DCIA = deep circumflex iliac artery flap; FFF = free fibula flap; FOM = floor of the mouth; G8 = Geriatric 8 screening tool; HADS = Hospital Anxiety and Depression Scale; HNC = head and neck cancer; KFI = Kaplan–Feinstein index; MRND = modified radical neck dissection; ND = neck dissection; NOS = not otherwise specified; OC = oral cavity; OCC = oral cavity cancer; OP = oropharynx; OOP = oral cavity and oropharynx; OOPC = oral/oropharyngeal cancer; ORFFF = osteofasciocutaneous radial forearm free flap; OSCC = oral squamous cell carcinoma; PMMC = pectoralis major myocutaneous flap; RFFF = radial forearm free flap; RT = radiotherapy; SCAIF = supraclavicular artery island flap; SCC = squamous cell carcinoma; SND = selective neck dissection; STSG = split thickness skin graft.

Article	Study Design	Country	Sample	Cohorts Definition	Independent Variables Considered (EORTC Questionnaires as Dependent Variable)	Findings
Airoldi, 2011 [22]	Cross-sectional study	Italy	38	OSCC undergoing RFFF and adjuvant RT	Other: dysphagia severity (grouping algorithm not clearly stated); psychological status (HADS)	Dysphagia severity: severe dysphagia group showed significantly worse global health status/QoL, fatigue, physical and social functioning, sexuality, social eating, and contacts Psychological status: depression showed positive correlation with poor head- and neck-specific functional domains (data not available)
Beck-Broichsitter, 2017 [24]	Cross-sectional study	Germany	50	OC undergoing surgery as primary treatment	Disease/treatment: T stage (Tis-2 vs. T3/4); mandibular involvement (no resection vs. marginal/segmental resection); reconstruction (local flaps NOS vs. distant flaps, including together PMMC, FFF, RFFF)	Reconstruction: local flaps group showed significantly better swallowing No statistical significance of other independent variables
Becker, 2012 [23]	Cross-sectional study	Germany	50	OC	Disease/treatment: site; T stage (Tis-2 vs. T3/4); mandibular involvement (no resection vs. marginal vs. segmental resection); reconstruction (not clearly reported)	Mandibular involvement: no resection showed significantly better results for all scales with the exception of cognitive functioning; marginal resection (compared to segmental resection) showed significantly better results for role functioning and financial difficulties T stage: early-stage group showed significantly better results in all scales; Reconstruction: “more invasive techniques” and combined reconstructions showed significantly worse results for role, emotional and social functioning, financial difficulties, pain, swallowing, speech problems, trouble with social eating, trouble with social contact No statistical significance of other independent variables
Borggreven, 2007 [25]	Prospective cohort study	The Netherlands	45	OOPC undergoing RFFF	Time (baseline vs. 6 months vs. 1 year) Sociodemographic: age; gender; marital status; comorbidity Disease/treatment: site (oral cavity vs. oropharynx); stage (T2 vs. T3-4); metachronous lesions/recurrence	Time: - Improvement at 6 months, preserved at 1 year for emotional functioning, insomnia, general and H&N pain, constipation - Late improvement observed only at 1 year for role functioning - Deterioration at 6 months, recovered to baseline levels at 1 year for physical function, social contacts, dental status - Deterioration at 6 months, partially recovered at 1 year for financial status, swallowing, social eating, dry mouth - Deterioration at 6 months, preserved at 1 year for senses, mouth opening, sticky saliva, coughing - Late deterioration observed only at 1 year for feeling ill - No statistical significance of other independent variables evaluated by linear regression
Bozec, 2009 [26]	Prospective Cohort study	France	50	OOPC undergoing RFFF without flap failure	Time (baseline vs. 6 months vs. 1 year) Sociodemographic: age; gender; comorbidity (KFI < 2 vs. ≥ 2) Disease/treatment: site (oral cavity vs. oropharynx); stage (AJCC2002 II vs. III/IV); RT	Time: significant progressive worsening of mouth opening from baseline to 6 and 1 year after treatment The statistical analysis on all sociodemographic and disease- and treatment-specific variables was performed on 6-month follow-up questionnaires and not considered for critical appraisal.
Bozec, 2020 [27]	Multicenter cross-sectional study	France	21	OOPC undergoing free flaps in elderly patients	Sociodemographic: age (<80 years vs. >80years); gender; educational level (< vs. ≥high school diploma); marital status/family (living at home alone vs. not); alcohol consumption (yes vs. no); tobacco consumption (yes vs. no) Disease/treatment: site (oral cavity vs. oropharynx); T stage (4 vs. <4); N stage (0 vs. >0); adjuvant RT Other: HADS (<15 vs. >15); Geriatric 8 health status scores (G8 < 15 vs. >15); number of patients concerns inventory (PCI)	HADS > 15 and G8 <15: significantly associated with poorer scores in global QoL score, functioning scales, general symptoms, H&N symptoms. The authors also administered the EORTC QLQ-ELD14 questionnaire, reporting significantly poorer results in patients older than 80 years, living alone, and with HADS > 15 in motility, as well as significantly poorer results in patients with HADS > 15 in joint stiffness, worries about the future, worries about others, burden of illness, maintaining purpose. Oropharyngeal cancers, G8 < 15 and HADS ≥ 15 were significantly associated with lower scores in the Dysphagia Outcome and Severity Scale (DOSS). HADS ≥ 15 has been significantly associated with a higher number of PCI. No statistical significance of other independent variables
Canis, 2016 [28]	Retrospective cohort study	Germany	48	Lateral tongue pT3 SCC primarily treated by surgical excision, neck dissection followed by CRT	Disease/treatment: reconstruction (RFFF vs. primary closure)	Reconstruction: RFFF group showed significant better speech, swallowing, and social eating
Crombie, 2014 [10]	Cross-sectional study	Australia	40	OC	Treatment by CRT alone vs. surgery alone/surgery with adjuvant RT/surgery with adjuvant CT	No statistically significant differences between compared groups
Davudov, 2019 [29]	Cross-sectional study	Iran	16	OCC undergoing mandible segmental resection	Disease/treatment: reconstruction (no reconstruction vs. free flap vs. plate)	Reconstruction: no reconstruction showed significantly worse outcomes in speech problems, dry mouth, and feeling ill
Dzioba, 2017 [30]	Prospective cohort study	Canada	120	Cancer of the anterior two-thirds of the tongue, treated by surgical excision and reconstruction alone or by a combination of surgery + RT or surgery + CRT	Time (baseline vs. 1 month vs. 6 months vs. 1 year) substratified by treatment protocol (surgery only vs. surgery + RT vs. surgery + CRT) only for some EORTC items	Surgery + RT group: - baseline vs. 1 year: significantly worse dry mouth at 1-year assessment Surgery + CRT group: - baseline vs. 1 year: significantly worse dry mouth at 1-year assessment - baseline vs. NOS: significantly worse results for eating, mouth opening, swallowing
Ferri, 2020 [31]	Multicenter retrospective cohort study	Italy	70	OSCC (T1-2, N0) involving the tongue and FOM undergoing transoral partial pelviglossectomy/BAMM flap or pull-through partial pelviglossectomy/free flap	Other: treatment protocol (transoral partial pelviglossectomy followed by BAMM flap vs. pull-through partial pelviglossectomy followed by free flap)	Significantly better results in transoral/BAMM flap group for average H&N35 questionnaire. The authors did not provide item-specific data, except for swallowing, which had significantly better result in the transoral/BAMM group
Girod, 2009 [32]	Prospective cohort study	USA	122	OC	Disease/treatment: reconstruction (ADM vs. STSG) substratified by RT (not specified if pre- or post-treatment); major complications (graft failure vs. regular healing)	Reconstruction: ADM group showed significantly better social eating Reconstruction stratified by RT: ADM/RT scored significantly better results in swallowing scale compared to STSG/RT No statistical significance of other independent variables
Huang, 2010 [33]	Cross-sectional study	Taiwan	41	HNC free from disease at least 2 y after combined treatment with curative intent	Sociodemographic: gender; age (32–48 years vs. 49–56 years vs. 57–83 years); marital status; educational level (≤6 years vs. 6–12 years vs. >12 years); family income (annual: <0.6 million NTD vs. 0.6–1.2 million NTD vs. ≥1.2 million NTD); comorbidity (Charlson Comorbidity Index [CCI]: 0 vs. ≥1) Disease/treatment: site (oral cavity vs. oropharynx vs. hypopharynx/larynx); stage (AJCC: II vs. III vs. IV); Other: treatment protocol (surgery + RT vs. surgery + RT + CT vs. RT + CT); RT dose (<63 Gy vs. ≥63 Gy); RT technique (2DRT vs. 3DCRT vs. IMRT); length of follow-up (2.2–3.5 years vs. 3.5–4.7 years vs. 4.7–13.2 years)	The study applied an interesting statistical model to compare several independent variables simultaneously in a double-step general linear model multivariate analysis of variance (GML-MANOVA). Annual family income: patients with ≥1.2 million NTD annual income showed significantly better results for physical functioning, role functioning, social functioning, financial problems, swallowing, speech, social eating, and social contact Site: - oral cancer patients showed significantly better results for physical functioning, cognitive functioning, fatigue, nausea/vomiting, pain, dyspnea, insomnia, appetite loss, pain, sense, speech, coughing, and feeling ill - hypopharyngeal/laryngeal cancer patients showed the worst results for mouth opening and coughing (statistically significant) - oropharyngeal cancer patients showed the worst results for dry mouth and sticky saliva (statistically significant) RT technique: patients treated by 3DCRT and IMRT showed significantly better results for swallowing, problems with teeth, mouth opening, dry mouth, and sticky saliva No statistically significant differences were found analyzing other independent variables (age, gender, educational level, marital status, comorbidity, cancer stage, RT dose, treatment protocol, length of follow-up)
Infante-Cossio, 2009 [34]	Prospective cohort study	Spain	67	OOPC	Time (baseline, 1 year, 3 years) Disease/treatment: site (oral cavity vs. oropharynx); adjuvant CRT Other: AJCC stage (I/II vs. III/IV)	Time: the study demonstrated three different evolution patterns among questionnaires items: (I) Improvement at the first and third year for emotional functioning, general pain, and specific H&N pain; (II) Worsening at the first year and improvement at the third year for global QoL, physical, role and social functioning, financial problems, sensory problems, social eating, social relationships, sexuality, mouth opening, and use of painkillers; (III) Worsening at the first and third year: cognitive functioning, fatigue, constipation, diarrhea, swallowing, speech, dry mouth, sticky saliva, cough, feeling ill, and weight loss. Site: oropharyngeal cancer showed worse results in overall QoL, functioning role, tiredness, nausea/emesis, appetite loss, pain, use of painkillers, dyspnea, social relationships Stage: III/IV stage cancers showed significantly worse state of health and QoL, pain, tiredness, loss of appetite, swallowing function, speech, social contacts, eating in public, mouth opening, cough, weight loss, use of pain killers Adjuvant CRT: patients undergoing adjuvant CRT showed significantly worse overall QoL, swallowing function, pain, dry mouth, sticky saliva, mouth opening, sensory disorders, speech, social eating
Kessler, 2004 [35]	Prospective cohort study	Germany	55	Primary OC undergoing nCRT + surgical excision or primary surgical excision + adjuvant RT	Time (baseline vs. 3 month vs. 1 year) substratified by treatment protocols (nCRT + surgery vs. surgery+RT)	nCRT + surgery group: - Baseline vs. 1 year: significantly worse results for global health status, physical function, role function, emotional function, social function, fatigue, nausea and vomiting, pain, dyspnea, insomnia, financial difficulties, swallowing, senses, speech, eating, social contact, teeth, mouth opening, dry mouth, sticky saliva, coughing, feeling ill, feeding tube Surgery + RT group: - Baseline vs. 1 year: significantly worse results for global health status, physical function, role function, emotional function, social function, fatigue, nausea and vomiting, pain, dyspnea, insomnia, financial difficulties, swallowing, senses, speech, eating, social contact, teeth, mouth opening, dry mouth, sticky saliva, coughing, feeling ill, feeding tube, weight gain
Khandelwal, 2017 [9]	Cross-sectional study	India	34	OC undergoing free flaps	Time (1–2 years vs. 3–5 years) Sociodemographic: age (<45 years vs. >45 years); gender Disease/treatment: site (anterior floor of the mouth/sublingual sulcus vs. retromolar region/tonsillar fossa/tongue); T stage (T2 vs. T3 vs. T4) Other: use of feeding tubes	T stage: progressively better results have been found for smaller tumors for global health status/QoL, functional scales, symptom scale, H and NSS (NOS). Feeding tubes: significantly worse results in patients using feeding tubes for functional status and H and N scales (NOS) No statistical significance of other independent variables
Klug, 2002 [36]	Retrospective cohort study	Austria	110	OC undergoing multimodal treatment (preoperative CRT followed by surgery and free flaps)	Disease/treatment: site (anterior vs. posterior); T stage (T2 vs. T4), mandibular involvement (segmental vs. marginal resection); neck dissection (SND vs. MRND (NOS)/bilateral ND)	No statistically significant differences between compared groups
Kovács, 2015 [37]	Cross-sectional study	Germany	100	OOPC undergoing various combinations of multimodality treatment	Sociodemographic: gender Disease/treatment: site (FOM vs. tongue vs. oropharynx vs. retromolar trigone vs. oral cheek vs. mandibular crest vs. lip vs. maxilla); neck dissection laterality (no vs. unilateral vs. bilateral) and type (super selective I-IIa vs. MRND-III); reconstruction (no vs. local flaps NOS vs. distant flaps NOS vs. free flaps NOS); adjuvant RT; adjuvant CRT; adjuvant CT. Other: time since treatment; comparison with EORTC group	Time since treatment: patients evaluated at the 4-years follow-up demonstrated statistically significant worse results for social eating and nutritional support compared to the 1-year follow-up evaluation. Gender: men showed significantly worse results for financial difficulties and cognitive and social functioning Site: cancers of the FOM showed significantly worse social contact compared to tongue; oropharyngeal cancers showed significantly worse results for feeding tubes and sticky saliva compared to tongue and retromolar trigone Reconstruction: - Distant flaps vs. free flaps: worse swallowing than free flaps in the former - Distant flaps vs. no reconstruction: worse swallowing, feeding tubes, social eating, and contact in the former - Distant flaps vs. local flaps: worse results for feeding tubes and social contact in the former - Free flaps vs. no reconstruction: worse need of feeding and social contact tube in the former - Free flaps vs. local flaps: worse results for social contact in the former Neck dissection: - Laterality: both unilateral and bilateral showed significantly worse results for mouth opening than no neck dissection group - Type: compared to super selective I-IIa and no neck dissection groups, MRNDIII showed significantly worse results for swallowing, speech, social eating and contact, sexuality, mouth opening, dry mouth, sticky saliva, feeding tubes, and weight loss Adjuvant therapy: - Adjuvant RT vs. no adjuvant therapy: the former showed significantly worse results for emotional and social functioning, appetite loss, swallowing, senses, speech, social eating and contact, sexuality, mouth opening, dry mouth, sticky saliva, pain, feeding tubes - Adjuvant CT vs. all other: the former showed significantly better results for sticky saliva - Adjuvant CRT showed the same results of adjuvant RT Comparison with the reference group: - Worse in studied sample: global health status, cognitive and social functioning, fatigue, social eating, dental status, mouth opening, dry mouth, and sticky saliva - Better in studied sample: H&N pain, need for pain killers, cough, need for nutritional support, weight loss and gain
Lin, 2020 [38]	Case control study	Taiwan	13	Cancer of the lower lip undergoing surgical resection and reconstruction with RFFF or barrel-shaped RFFF	Disease/treatment: reconstruction (RFFF vs. barrel-shaped RFFF)	Reconstruction: patients undergone barrel-shaped RFFF reconstruction scored better results for swallowing, speech, social eating, social contact and dry mouth
Mair, 2017 [39]	Prospective cohort study	India	38	T4 cancers of the buccal mucosa undergoing surgery (ablation, neck dissection and reconstruction with PMMC) as first-line treatment	Time (baseline vs. 3 months vs. 6 months vs. 9 months vs. 1 year) on the disease-free sub cohort and sub stratified by adjuvant therapy Disease/treatment: adjuvant therapy (RT vs. CRT)	Baseline differences between disease-free patients and those who developed a relapse: significantly worse results in the latter group for global QOL, dyspnea, appetite loss and weight loss Adjuvant therapy: no differences at 1-year evaluation between groups
Moubayed, 2014 [40]	Cross-sectional study and systematic review of literature	Canada	37	OSCC undergoing segmental resection of the mandible and free flaps	Disease/treatment: reconstruction (FFF vs. ORFFF vs. Scapular flap)	No statistically significant differences between compared groups
Nordgren, 2008 [41]	Multicenter prospective cohort study	Sweden/Norway	37	OC	Time (baseline vs. 3 months vs. 6 months vs. 1 year vs. 5 years) in entire cohort and substratified by treatment protocol and survival Other: treatment protocol (surgery alone vs. RT alone vs. combined); survival (5-year survivors vs. 5-year non-survivors and 5-year survivors vs. died after the first year)	Time (baseline vs. 5 years) entire cohort: significant improvement in emotional functioning, significant deterioration in physical and role functioning, dyspnea, problems with senses, teeth, mouth opening, dry mouth, and sticky saliva Time (1 year vs. 5 years) entire cohort: significant deterioration in role functioning, sticky saliva, and mouth opening Time (baseline vs. 5 years) surgery alone: stability of all items Time (baseline vs. 5 years) RT alone: significant improvement of sleep disturbance, H&N pain, social eating and mouth opening; deterioration in physical and role functioning, dyspnea, senses, and dry mouth. Time (baseline vs. 5 years) combined group: significant improvement for emotional functioning and sleep problems; deterioration for role functioning, senses, mouth opening, dry mouth, and sticky saliva. 5-year survivors vs. 5-year non-survivors (compared at baseline): survivors showed significantly better results at baseline for physical, cognitive, and social functioning; fatigue; pain; dyspnea; sleep disturbance; appetite loss; H&N pain; senses; speech; social eating and contacts; dental status; mouth opening; sticky saliva; and dry mouth 5-year survivors vs. died after the first year (compared at baseline): survivors showed significantly better results for physical, cognitive, and social functioning; fatigue; pain; dyspnea; sleep disturbance; appetite loss; H&N pain; senses; speech; social eating; dental status; mouth opening; dry mouth; sticky saliva 5-year survivors vs. died after the first year (compared at 1 year): survivors showed significantly better results for physical and role functioning, fatigue, nausea/vomiting, appetite loss, constipation, diarrhea, swallowing, social eating, sexuality, mouth opening.
Oates, 2008 [42]	Prospective cohort study	Australia	47	HNC	Time (baseline vs. 3 months vs. 6 months vs. 1 year) substratified by site and treatment protocol Disease/treatment: site (oral cavity vs. oropharynx vs. larynx vs. nasopharynx vs. parotid vs. occult primary vs. paranasal sinus) Other: treatment protocol (surgery vs. RT only) substratified by site	Patients undergoing RT only over time: - oral cavity: significant improvement in emotional functioning over time - oropharynx: significant deterioration of dry mouth over time - larynx: significant improvement in emotional, cognitive, and social functioning
Oskam, 2013 [43]	Prospective cohort study	The Netherlands	129	OOPC	Time (baseline vs. 6 months vs. 1 year vs. ≥8 years) Sociodemographic: age (NOS); gender; marital status Disease/treatment: tumor site (OC vs. OP); stage (NOS) Other: long-term survival	Time: the mixed-effects model showed significant deterioration from baseline to long-term evaluation for dry mouth, sticky saliva, speech, coughing, senses, swallowing, and social functioning. Long-term survival: non-survivors showed significantly worse baseline global health status/QoL, general pain, appetite loss, swallowing, dental status, and feeling ill No statistical significance of other independent variables
Peisker, 2016 [44]	Cross-sectional study	Germany	22	OSCC undergoing free flaps	None	Authors performed a bivariate intraquestionnaire analysis to correlate impact of symptom scales on global health status/QoL scale
Petruson, 2005 [45]	Prospective cohort study	Sweden	225	Primary OOPC (mobile tongue vs. OPC) undergoing brachytherapy	Time (baseline vs. 3 months vs. 1 year vs. 3 years) substratified by site (mobile tongue vs. OPC), brachytherapy quality indices dose, dose rate, and tumor target volume	Mobile tongue group: - baseline vs. 1 year post-treatment: significantly worse dry mouth - baseline vs. 3 years post-treatment: significantly worse dry mouth - brachytherapy dose rate: significant association NOS between brachytherapy dose rate and swallowing solid food at NOS timepoint
Pierre, 2014 [46]	Prospective cohort study	France	117	OOPC undergoing free flaps without flap failure and disease free	Sociodemographic: age (>70 years vs. <70 years); gender; comorbidity (KFI ≥2 vs. <2); Disease/treatment: site (oral cavity vs. oropharynx) and OOP subsites (mobile tongue vs. FOM vs. cheek vs. hard palate vs. BOT vs. pharyngeal wall vs. soft palate vs. posterior pharyngeal wall); T stage (T2 vs. T3 vs. T4); mandibular involvement (no vs. segmental resection); reconstruction (FFF/scapular vs. RFFF/ALT); adjuvant RT; neoadjuvant RT; N stage (N ≥ 1 vs. N0)	T stage: T3–4 stage group showed significantly worse results in mean QoL global score, mean C30 symptom domains score and mean H&N35 module score Subsite: BOT showed a significantly worse result in mean H&N35 module score Adjuvant RT: significantly worse results in mean H&N35 module score Neoadjuvant RT: significantly worse results in mean H&N35 module score No statistical significance of other independent variables
Schoen, 2008 [47]	Prospective cohort study	The Netherlands	41	OOPC in edentulous undergoing surgical excision and implant retained prosthesis rehabilitation	Time (baseline vs. 6 weeks vs. 1 year) substratified by adjuvant RT Disease/treatment: adjuvant RT	Adjuvant RT: patients undergoing adjuvant radiotherapy showed significantly worse results for H&N pain, swallowing, speech, social eating, sexuality, mouth opening, dry mouth, and sticky saliva. Significantly better result was shown in nausea/vomiting.
Van Gemert, 2015 [48]	Cross-sectional study	The Netherlands	20	OC undergoing lateral segmental resection of the mandible	Sociodemographic: age (NOS); gender Disease/treatment: site (retromolar area vs. FOM vs. gingiva vs. cheek); neck dissection (no vs. unilateral NOS vs. bilateral NOS); reconstruction (of the bony defect [FFF vs. plate] and of soft tissue defect among plate group [primary closure vs. RFFF vs. PMMC]); adjuvant RT Other: cN stage (0 vs. +); horizontal defect size; occlusion (achieved vs. not achieved); accessory nerve sacrifice	Age: significant inverse relation with mouth opening (OVB or selection bias) Gender: relation NOS with feeding tube (OVB or selection bias) Reconstruction of the bony defect: significant relation NOS with functional scales and feeling ill Reconstruction of soft tissue defect: significant relation NOS with mouth opening and feeling ill Bilateral neck dissection NOS: significant relation NOS with social eating and contact, dental status, and feeding tube Horizontal defect size: significant relation NOS with feeding tube Accessory nerve sacrifice: significant relation with swallowing and speech troubles No statistical significance of other independent variables
Yoshimura, 2009 [49]	Prospective cohort study	Japan	30	OC undergoing primary low-dose-rate brachytherapy with no cervical lymph node or distant metastases, no other active malignancies	Time (baseline vs. 3 months vs. 6 months vs. 1 year) Sociodemographic: gender; age (<65 years or >65 years) Disease/treatment: site (tongue vs. others); T stage (T1 vs. T2–3) Other: brachytherapy source (iridium vs. cesium vs. gold)	Site: patients affected by cancer of the tongue scored worse results at baseline for swallowing, senses and sticky saliva. The latter two remained worse during the follow-up period (1 y), while swallowing item improved toward results comparable with those of the other group at 1 y assessment T stage: T1 stage patients demonstrated higher scores for global health status at baseline and at the 1-year evaluation No statistical significance of other independent variables
Tot			1833			

## Data Availability

The data presented in this study are available on request from the corresponding author.

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
