# Peer review of "Health-Related Quality of Life in Oral Cancer Patients: Scoping Review and Critical Appraisal of Investigated Determinants"

_cancers, 2021, doi:10.3390/cancers13174398_

Round 1
Reviewer 1 Report
The idea is original, the methods were in general appropriate, figures, tables and schemes excellent, and the conclusions harbor translational opportunities and recommendations. This is an important paper that in my opinion should be considered for publication in Cancers.
My following constructive comments are suggested:
- Title: According to PRISMA statement for scoping reviews (i.e., Preferred Reporting Items for Systematic reviews and Meta-Analyses extension for Scoping Reviews (PRISMA-ScR) Checklist), the study was identified as a Scoping review (item 1: Identify the report as a scoping review.).
But, why as a “PRISMA” scoping review? PRISMA are reporting guidelines, it is a frequent controversial point I usually observe in systematic reviews and meta-analyses e.g., a PRISMA compliant systematic review…I strongly suggest to delete this term from title, it was only a reporting approach, already commented in the material and methods section.
- “A systematic search of published literature was performed in PubMed, EMBASE and 87 Scopus databases (last screening on February 2, 2021)”. Please, it should be clarified if a lower date limit was used in search strategy.
- Thesaurus terms were not used in search strategy, as highly recommended (e.g., Mesh for MEDLINE/PubMed and/or Emtree for Embase). It should be clarified and/or justified.
- I understood that non-English language published studies were excluded. It the most critical point that should be clarified and/or corrected. Language bias is probably one of the most important considered selection biases in systematic reviews. The sample size was optimal (n=30), but 929+440+198 studies were respectively excluded after language filters, titles/abstracts screening and full-text reading. The quality of evidence offered by the outcomes evaluated in this scoping review could be seriously downgraded.
My proposal: The number of studies excluded should be reported as supplementary information with their exclusion reasons and references. Then, the number of studies excluded by language -and meeting the rest of the inclusion criteria- should be re-considered to be included and analyzed in this scoping review to avoid a potential language bias. This will ensure that the results of the studies not published in the English language truly represent - or differ from - the results of the studies included in this first version of the manuscript, better covering the literature, reducing the risk of publication bias and increasing the quality of the evidence for the outcomes analyzed.
Reviewer 2 Report
Dear Authors,
English language and style are fine.
Punctuation and editorial errors in the text should be corrected.
Materials and methods
Add figure - PRISMA flow diagram of study selection (no in results)
Table 1 - the caption should be above the table.
Table 2 - the caption should be above the table.
Table 3 - the caption should be above the table.
In English, we write numbers after the dot, e.g. 3,7 vs. 3.7
line 461 4.2.4. Extent of resection skip to page 30
The quality of the figures is too low (e.g. Figure 1, 2).
Incorrect citation record type at reference point.
Apolone G, De Carli G, Brunetti M, Garattini S. Health-related quality of life (HR-QOL) and regulatory issues. An assessment of the European Agency for the Evaluation of Medicinal Products (EMEA) recommendations on the use of HR-QOL measures in drug approval. PharmacoEconomics. 2001;19(2):187-95. 10.2165/00019053-200119020-00005
vs.
Zhang, J.; Yu, X.; Guo, P.; Firrman, J.; Pouchnik, D.; Diao, Y.; Samulski, R.J.; Xiao, W. Satellite Subgenomic Particles Are Key Regulators of Adeno-Associated Virus Life Cycle. Viruses 2021, 13, 1185.
To sum up, article can be accepted after major revision.
Round 2
Reviewer 1 Report
All comments have been taken into consideration. The paper is now suitable for publication.
Reviewer 2 Report
Dear Authors,
Thanks for your corrections.
"All references have been controlled and eventual errors have been corrected, but we have not cited the article by Zhang J et Al.'
I included this reference as an example: Zhang, J.; Yu, X.; Guo, P.; Firrman, J.; Pouchnik, D.; Diao, Y.; Samulski, R.J.; Xiao, W. Satellite Subgenomic Particles Are Key Regulators of Adeno-Associated Virus Life Cycle. Viruses 2021, 13, 1185.
The authors did not correct the references properly.
114 references need to be improved !!!
Accept after minor revision.